# Apigenin Ameliorates H_2_O_2_-Induced Oxidative Damage in Melanocytes through Nuclear Factor-E2-Related Factor 2 (Nrf2) and Phosphatidylinositol 3-Kinase (PI3K)/Protein Kinase B (Akt)/Mammalian Target of Rapamycin (mTOR) Pathways and Reducing the Generation of Reactive Oxygen Species (ROS) in Zebrafish

**DOI:** 10.3390/ph17101302

**Published:** 2024-09-30

**Authors:** Qing-Qing Tang, Zu-Ding Wang, Xiao-Hong An, Xin-Yuan Zhou, Rong-Zhan Zhang, Xiao Zhan, Wei Zhang, Jia Zhou

**Affiliations:** 1School of Traditional Chinese Pharmacy, China Pharmaceutical University, Nanjing 211198, China; tqq697752@163.com (Q.-Q.T.); zhouxinyuan0909@163.com (X.-Y.Z.); zrz758910@163.com (R.-Z.Z.); 15621350704@163.com (X.Z.); 2Yunnan Characteristic Plant Extraction Laboratory, Yunnan Yunke Characteristic Plant Extraction Laboratory Co., Ltd., Kunming 750021, China; 15987130746@163.com (Z.-D.W.); 15298366363@163.com (X.-H.A.); 3Institute of Dermatology, Chinese Academy of Medical Sciences and Peking Union Medical College, Nanjing 210042, China

**Keywords:** apigenin, melanogenesis, melanocyte dendricity, PINK1/Parkin signal pathway, PI3K/Akt/mTOR signal pathway

## Abstract

**Background:** Apigenin is one of the natural flavonoids found mainly in natural plants, as well as some fruits and vegetables, with celery in particular being the most abundant. Apigenin has antioxidant, anti-tumor, anti-inflammatory, and anticancer effects. In this research, we attempted to further investigate the effects of apigenin on the mechanism of repairing oxidative cell damage. The present study hopes to provide a potential candidate for abnormal skin pigmentation disorders. **Methods:** We used 0.4 mM H_2_O_2_ to treat B16F10 cells for 12 h to establish a model of oxidative stress in melanocytes, and then we gave apigenin (0.1~5 μM) to B16F10 cells for 48 h, and detected the expression levels of melanin synthesis-related proteins, dendritic regulation-related proteins, antioxidant signaling pathway- and Nrf2 signaling pathway-related proteins, autophagy, and autophagy-regulated pathways by immunoblotting using Western blotting. The expression levels of PI3K/Akt/mTOR proteins were measured by β-galactosidase staining and Western blotting for cellular decay, JC-1 staining for mitochondrial membrane potential, and Western blotting for mitochondrial fusion- and mitochondrial autophagy-related proteins. **Results:** Apigenin exerts antioxidant effects by activating the Nrf2 pathway, and apigenin up-regulates the expression of melanin synthesis-related proteins Tyr, TRP1, TRP2, and gp100, which are reduced in melanocytes under oxidative stress. By inhibiting the expression of senescence-related proteins p53 and p21, and delaying cellular senescence, we detected the mitochondrial membrane potential using JC-1, and found that apigenin improved the reduction in mitochondrial membrane potential in melanocytes under oxidative stress, and maintained the normal function of mitochondria. In addition, we further detected the key regulatory proteins of mitochondrial fusion and division, MFF, p-DRP1 (S637), and p-DRP1 (S616), and found that apigenin inhibited the down-regulation of fusion-associated protein, p-DRP1 (S637), and the up-regulation of division-associated proteins, MFF and p-DRP1 (S616), due to oxidative stress in melanocytes, and promoted the mitochondrial fusion and ameliorated the imbalance between mitochondrial division and fusion. We further detected the expression of fusion-related proteins OPA1 and Mitofusion-1, and found that apigenin restored the expression of the above fusion proteins under oxidative stress, which further indicated that apigenin promoted mitochondrial fusion, improved the imbalance between mitochondrial division and fusion, and delayed the loss of mitochondrial membrane potential. Apigenin promotes the expression of melanocyte autophagy-related proteins and the key mitochondrial autophagy proteins BNIP3L/Nix under oxidative stress, and activates the PINK1/Parkin signaling pathway by up-regulating the expression of autophagy-related proteins, as well as the expression of PINK1 and Parkin proteins, to promote melanocyte autophagy and mitochondrial autophagy. **Conclusions:** Apigenin exerts anti-melanocyte premature aging and detachment effects by promoting melanin synthesis, autophagy, and mitochondrial autophagy in melanocytes, and inhibiting oxidative cell damage and senescence.

## 1. Introduction

Apigenin is a typical flavonoid that is mainly found in natural plants such as Asteraceae and Chamomile. Some fruits and vegetables, such as grapefruit and celery, also contain apigenin. Studies have shown that apigenin has pharmacological activities such as hypotensive [1], anti-inflammatory [2], antioxidant [3], and anticancer [4]. Therefore, they have the advantages of better taste and high nutritional value. Celery is rich in apigenin, which is esterified with the resulting carboxylic acid during frying, thereby increasing cellular antioxidant activity [5]. Apigenin also regulates non-intrinsic apoptotic pathways through the activation of cysteine protease 8 (caspase8). In tumor cells, apigenin activates apoptosis by regulating the expression of Bcl-2, Bax, STAT3, and Akt proteins [6]. Apigenin also inhibits the phosphorylation of JAK2, STAT3, and STAT5, thereby inhibiting the metastasis of cancer cells. Inhibition of STAT3 phosphorylation resulted in down-regulation of the expression of STAT3 target proteins, including matrix metallopeptidases MMP-2 and MMP-9, vascular endothelial growth factor (VEGF), and twist-associated protein 1 (Twist1), which play key roles in cancer cell migration and invasion [7,8]. Researchers found that apigenin can significantly increase tyrosinase content and promote melanin synthesis. However, the associated mechanisms have mostly remained unknown [9].

Melanin is mainly produced and secreted by melanocytes present in the spiny layer of the skin [10]. Melanin is mainly synthesized, stored, and transported in melanosomes. The maturation of melanosomes is divided into four stages: I, II, III, and IV. Stage Ⅰ melanosomes are a class of endosomes with a structure similar to that of internal vesicles, which contain an amorphous matrix, no pigment deposition, and the premelanosome protein (PMEL) begins to assemble in them. In Stage II, the fibronectin PMEL has been synthesized in the melanosomes, and enzymes and other proteins involved in melanin synthesis are translocated into the melanosomes. In Stages III and IV, melanin is synthesized and completely deposited in the melanosomes [10,11,12]. The key enzymes that affect melanin production are tyrosinase (TYR) and tyrosine-related proteins 1 and 2 (TRP1 and TRP2); among them, TYR is the main rate-limiting enzyme, which determines the rate and production of melanin production [13]. Tyrosinase is converted to dopaquinone (DQ) by the catalytic action of tyrosinase. In melanosomes, DQ interacts with sufficient cysteine to eventually convert to brown melanin, which is then catalyzed by a multienzyme complex formed by the TRP1 and TRP2 that facilitates the stabilization of tyrosinase to eventually form true melanin [11].

Vitiligo, an acquired, progressive depigmentation of the skin and mucous membranes, is characterized by leucoderma-like lesions caused by the destruction of the structure and function of melanocytes, and the lesions can appear on the head, face, hands, feet, and other parts of the body [14,15]. Vitiligo is a pigmented disease, and its main pathological feature is manifested as a reduction or even disappearance of the number of melanocytes in the lesion area. Regarding the pathogenesis of vitiligo, scholars have put forward a variety of hypotheses, such as autoimmunity, oxidative stress, and genetic variations [16]. Among these hypotheses, oxidative stress-induced melanocytotoxicity and a reduced number of melanocytes due to melanocyte detachment have been recognized by many scholars. Oxidative stress is an imbalance between oxidation and reduction in cells, resulting in a stress response triggered by the accumulation of reactive oxygen species (ROS), which can damage DNA and oxidize a variety of DNA repair proteins, reducing their efficiency, interfering with melanin synthesis, and leading to melanocyte apoptosis [17,18]. Oxidative stress plays an important role in the development and course of vitiligo. According to research, vitiligo patients have significant levels of oxidative stress, with an imbalance between oxidation and reduction, an excessive accumulation of ROS causing melanocyte damage and apoptosis, cellular dysfunction, and a reduced melanin synthesis [15,19].

The Nrf2/ARE antioxidant pathway is one of the most important cellular defense mechanisms against oxidative stress damage, and plays an important role in ameliorating the process of melanocyte damage by oxidative stress [20]. Nrf2 (Nuclear factor-E2-related factor 2) is a member of the CNC (Cap-N-Collar) family of transcription factors, which can regulate the expression of a series of antioxidant factors, such as GCL, HO-1, and NQO1 [21]. Under normal physiological conditions, Nrf2 binds to the cytoplasmic chaperone protein Keap1 in a relatively inhibited state and is in a quiescent state; when subjected to oxidative stimuli, Nrf2 and Keap1 are depolymerized and phosphorylated by a variety of protein kinases and then transferred into the nucleus, where they bind to the relevant proteins in the nucleus and proceed to recognize and bind to the ARE genes, which then activate the downstream antioxidant-associated proteins, exerting a powerful antioxidant effect [22]. The present study focuses on the repigmentation effect of apigenin on vitiligo lesions, and whether it plays an antioxidant role by activating the Nrf2/ARE pathway, so as to maintain the normal physiological state and function of melanocytes.

## 2. Results

### 2.1. Establishment of an Oxidative Stress Injury Model for B16F10 Melanocytes

Oxidative damage models are commonly induced by H_2_O_2_. We first performed in vitro experiments. As shown in Figure 1A, apigenin significantly reduced the elevation of ROS caused by oxidative stress in zebrafish. In order to determine the effects of different concentrations of H_2_O_2_ on cell viability and intracellular ROS production, B16F10 cells were treated with various concentrations of H_2_O_2_ (0~1 mM) for a duration of 12 h. The effect of H_2_O_2_ on the cell viability was detected by MTT assay. As shown in Figure 1B, there was less cytotoxicity in the concentration range of H_2_O_2_ at 0.1~0.4 mM, which was selected for further experiments. And then, we examined the effect of H_2_O_2_ on ROS generation in B16F10 cells with the ROS assay. The results showed that 0.4 mM H_2_O_2_ caused less cellular damage and significantly increased the intracellular ROS levels (Appendix A); therefore, 0.4 mM H_2_O_2_ for 12 h was finally chosen to fabricate the oxidative stress model of B16F10 cells.

### 2.2. Effects of Apigenin on ROS Level in B16F10 Cells

The intracellular scavenging mechanisms of free radicals include two main types of antioxidant enzymes and small molecule antioxidants. Antioxidant enzymes include catalase, superoxide dismutase, glutathione peroxidase, glutathione reductase, and glyoxalase. As shown in Figure 1C,D, we found that apigenin could not only significantly reduce the accumulation of ROS in B16F10 cells, but also increase the activity of the antioxidant enzyme SOD and reduce the production of MDA (Figure 2A,B). As a conclusion, apigenin can attenuate oxidative damage caused by oxidative stress in B16F10 cells.

### 2.3. Apigenin Inhibited ROS Production Caused by Oxidative Stress via Nrf2/ARE

There is a significant link between Nrf2 and the pathogenesis of vitiligo. Nuclear transcription factors such as Nrf2 are essential for controlling oxidative stress in cells. When under oxidative stress, Nrf2 is translocated from the cytoplasm to the nucleus, where it binds to antioxidant response elements (AREs) located in the promoters of genes encoding antioxidant enzymes and controls the expression of related antioxidant enzymes (HO-1, NQO1, and SOD), thus exerting antioxidant effects. Therefore, we examined the effect of apigenin on the intracellular antioxidant Nrf2 signal pathway. The Western blot results showed that apigenin increased the expression of Nrf2, HO-1, and NQO1 proteins under oxidative stress (Figure 2C–G). Keap1, a part of the E3 ubiquitin ligase, ubiquitylates Nrf2, which is degraded by the proteasome, and the results indicated that apigenin can inhibit Keap1 expression and maintain Nrf2 stability. The above experimental results indicate that apigenin can activate the Nrf2/ARE signal pathway.

### 2.4. Effect of Melanocyte Senescence Caused by Oxidative Stress after Treatment with Apigenin

The β-galactosidase staining results showed that H_2_O_2_ significantly increased β-galactosidase activity in melanocytes, but apigenin pretreatment significantly inhibited the increase in the β-galactosidase activity induced by oxidative stress in the melanocytes (Figure 3A). The Western blot analysis was used to investigate the effects of apigenin on the levels of p53 and p21 proteins’ expression. The results demonstrated that cells treated with H_2_O_2_ showed an increase in the levels of p53 and p21 expression, while these changes were attenuated by apigenin (Figure 3B–D). This suggests that apigenin has the ability to attenuate the β-galactosidase activity and the p53 and p21 proteins’ expression in a dose-dependent manner, thus delaying cellular senescence.

### 2.5. Effects of Apigenin on Melanin Synthesis in Oxidant B16F10 Cells

To investigate the effects of apigenin on melanin synthesis affected by oxidative stress, we examined the levels of TYR, TRP1, TRP2, and gp100 proteins’ expression. As shown in Figure 4A–E, apigenin could increase the levels of TYR, TRP1, and TRP2 proteins’ expression in a dose-dependent manner. Furthermore, apigenin alleviated the inhibition of melanosome protein gp100 expression caused by oxidative stress. According to the findings, it is suggested that apigenin could regulate melanogenesis by increasing the inhibition of the expression of TYR, TRP1, TRP2, and gp100 induced by oxidative stress.

### 2.6. Effect of Apigenin on Cytoskeleton of B16F10 Cells in Oxidative Stress States

In order to examine whether apigenin can improve the atrophy of dendrites or pseudopods in oxidative damage melanocytes, we determined cytoskeletal changes by FITC–Phalloidin fluorescent probe staining and analyzed the expression of dendritic pseudopod regulation and adhesion-related proteins by Western blotting. In our investigation, it was observed that apigenin effectively enhanced the alleviation of dendrite atrophy induced by oxidative stress in melanocytes. Additionally, apigenin exhibited a significant influence on augmenting the elongation and quantity of F-actin (Figure 4F). In addition, Western blot results showed that 5 µM apigenin alleviated the suppression of Rac-1 and Cdc42 proteins’ expression caused by oxidative stress, and increased the expression level of E-Cadherin protein (Figure 4G–J). E-Cadherin is a class of membrane proteins that mediate cell adhesion and sensing of external information.

### 2.7. Effect of Apigenin on Mitochondria under Oxidative Stress in Melanocytes

Mitochondria are organelles closely related to energy metabolism. Mitochondria can maintain the intracellular energy metabolism and homeostasis of the internal environment; once the mitochondria are damaged, the cell will lose vitality, and the mitochondrial membrane potential (MMP) that maintains the normal function of mitochondria plays an important role; it is an indispensable part of the process of cellular oxidative phosphorylation to generate energy. The decrease in MMP is also a sign of early apoptosis. When the MMP is higher, JC-1 aggregates and forms polymers in the mitochondrial matrix, resulting in red fluorescence; on the contrary, when the MMP is lower, JC-1 is unable to aggregate in the mitochondrial matrix and produces green fluorescence. The JC-1 staining results showed that the intensity of red fluorescence decreased, and green fluorescence increased significantly in B16F10 cells after H_2_O_2_ treatment compared with the control group, indicating that the mitochondria were damaged and dysfunctional. Compared with the model group, apigenin pretreatment effectively ameliorated the MMP decrease that was induced by oxidative stress (Figure 5A). The results indicated that apigenin protected the MMP, and enhanced the mitochondrial activity, which had a protective effect on the mitochondrial disorder in H_2_O_2_-treated B16F10 cells.

### 2.8. Effect of Apigenin on Mitochondrial Fusion and Mitochondrial Fission

Mitochondrial fusion and fission are a dynamically balanced process within the cell and play a critical role in maintaining normal mitochondrial function. Therefore, we examined the key proteins’ expression during mitochondrial fusion and fission. As shown in Figure 5B–G, the expression of the fusion-associated protein (OPA1, Mitofusion-1) and the fission-associated protein p-DRP1(S637) were suppressed, and the p-DRP1(S616) and MFF proteins’ expression were increased in B16F10 cells under oxidative stress, indicating that mitochondrial fission was increased, and fusion was decreased. Apigenin could promote the expression of the above fusion proteins in B16F10 cells and p-DRP1(S637), and inhibit p-DRP1(S616) and MFF expression. These results suggest that apigenin promotes mitochondrial fusion, improves the imbalance between mitochondrial fusion and fission, improves mitochondrial membrane potential, and inhibits apoptosis in B16F10 cells under oxidative stress.

### 2.9. Effects of Apigenin on the Levels of Relating Autophagic Proteins of B16F10 Cells

As shown in Figure 5H–N, the proteins of Beclin-1, Atg5, Atg8, Atg12, LC3-Ⅰ/II, and p62 expression were inhibited after H_2_O_2_ treatment in cells. Apigenin pretreatment, on the other hand, alleviated autophagy inhibition and dose-dependently promoted the levels of autophagy-related proteins’ expression. The results indicated that oxidative stress inhibits autophagy in B16F10 cells, and apigenin can reduce the oxidative damage of melanocytes under oxidative stress by promoting cellular autophagy, maintaining the normal state and function of cells, and reducing apoptosis.

### 2.10. Effects of Apigenin on the Levels of Mitophagy Relating Proteins of B16F10 Cells

Mitochondrial autophagy is the removal of senescent and damaged mitochondria by cells through selective autophagy. Therefore, we examined the levels of PINK1, Parkin, and BNIP3L/Nix proteins’ expression during mitochondrial autophagy. As shown in Figure 6A–D, after H_2_O_2_ treatment, the cells showed reductions in BNIP3L/Nix, PINK1, and Parkin expression. Apigenin (5 µM) pretreatment rescued the expression of BNIP3L/Nix, PINK1, and Parkin proteins after H_2_O_2_ treatment. These results suggested that apigenin might promote mitochondrial autophagy in B16F10 cells by activating PINK1/Parkin pathways.

### 2.11. Effects of Apigenin on Levels of PI3K/Akt/mTOR Signal Pathway in B16F10 Cells

The PI3K/Akt/mTOR pathway is known to be a pathway that regulates autophagy and is a negative regulator of autophagy. With activation of this pathway, cellular autophagy is inhibited. Therefore, we tested the effect of apigenin on the PI3K/Akt/mTOR pathway in B16F10 cells under oxidative stress. We detected PI3K, p-Akt, Akt, p-mTOR, and mTOR proteins’ expression by Western blot. The results showed that the above proteins’ expression was increased by oxidative stress, whereas apigenin or PI3K inhibitor (LY294002) inhibited the expression of the above proteins in the state of oxidative stress, and the cotreatment of both of them enhanced the inhibitory effect further. In addition, we found that apigenin and rapamycin (RAPA) inhibited mTOR phosphorylation and p62 up-regulation in B16F10 cells, and again, the inhibitory effect was enhanced by the combination of them. The above results suggest that apigenin inhibited the PI3K/Akt/mTOR pathway to reduce the promotion of ROS and then promoted autophagy in B16F10 cells under oxidative stress (Figure 6E–K).

## 3. Discussion

Apigenin is one of the natural flavonoids mainly found in natural plants such as Asteraceae, Rafflesiaceae, Verbenaceae, and some fruits and vegetables, especially celery, with the highest content. Apigenin from natural sources has high safety and low toxicity. Currently, research on apigenin focuses on cardiovascular protection and antioxidant, anti-tumor, anti-inflammatory, anticancer, and more effects [2,3,4]. One study showed that apigenin significantly promoted melanogenesis and the tyrosinase activity of B16F10 cells [9].

Oxidative stress is an unavoidable state of the organism, in which the organism generates free radicals such as the antioxidant enzymes glutathione peroxide, superoxide dismutase, and catalase. Under normal circumstances, the generation and elimination of oxygen free radicals in the body are in a state of dynamic equilibrium [23]. Some studies have shown that oxidative stress is an important causative factor of vitiligo. Melanocytes in the state of oxidative stress undergo melanin loss, dendritic atrophy, and reduced adhesion, which leads to premature senescence and loss of melanocytes, and, ultimately, the formation of milky-white patches typical of vitiligo on the skin [24].

Nrf2, a class of nuclear transcription factors that regulates redox homeostasis in vivo to protect against oxidative damage, can regulate the expression of related antioxidant enzymes (SOD) through the activation of the AREs, thus exerting antioxidant effects [25]. HO-1 and NQO1 are important antioxidant proteases that protect cells against oxidative damage [26]. Keap1 is a regulatory protein of Nrf2, which ubiquitinates Nrf2 and is thus degraded by the proteasome [27]. The experimental results showed that apigenin inhibited the Keap1 and increased Nrf2, HO-1, and NQO1 proteins’ expression to protect against oxidative damage. Furthermore, β-galactosidase is a cellular senescence biomarker; β-galactosidase is commonly used to identify senescent cells as an important biological sign of cellular senescence. In senescent cells, β-galactosidase activity is significantly elevated [28]. p21 is a cell cycle-inhibitory protein that acts as the downstream of the p52 signal molecule, and plays an important role in cell cycle arrest, cell differentiation, and apoptosis [29,30]. Experimental results show that oxidative stress increases β-galactosidase activity and promotes p53 and p21 proteins’ expression in B16F10 cells. These results suggest apigenin could activate the Nrf2 pathway and inhibit p53 and p21 proteins’ expression to ameliorate oxidative stress damage, melanocyte detachment, and senescence (Figure 1, Figure 2 and Figure 3).

The results of FITC–Phalloidin fluorescent probe staining showed that H_2_O_2_ treatment led to dendritic atrophy and a reduction in dendritic number in B16F10 cells; the results of Western blotting showed that H_2_O_2_ treatment led to a reduction in the expression of melanin synthesis, and transport-related proteins TYR, TRP1, TRP2, gp100, Rac-1, Cdc42, and E-Cadherin proteins’ expression was reduced; however, apigenin treatment was able to ameliorate the cellular dendritic atrophy and the reduction in dendritic number caused by oxidative stress in B16F10 cells, and it was also able to up-regulate the expression of melanin synthesis and transport-related proteins, which could improve the oxidative stress injury in B16F10 cells, promote the synthesis of melanin, improve the cellular dendritic atrophy, and promote melanocyte adhesion (Figure 4).

Mitochondria are one of the important sources of ROS in the cell. Damage to mitochondria leads to an increase in ROS content, and ROS production directly activates the mitochondrial permeability transition, leading to an MMP decrease [31]. We found that apigenin ameliorated the production of ROS and the decrease in membrane potential in melanocytes under oxidative stress, thereby maintaining normal mitochondrial function and inhibiting apoptosis (Figure 1C,D and Figure 5A). Mitochondrial fission and fusion are dynamically balanced, which is a crucial requirement for ensuring membrane structural integrity and an important mechanism for maintaining normal membrane potential, which is critical for mitochondrial activity. We found that apigenin improved the inhibition of p-DRP1(S637) expression, decreased the DRP1, p-DRP1(S616), and MFF expression, and increased the expression of mitochondrial fusion protein, OPA1, and Mitofusion-1 (Figure 5B–G). These findings imply that apigenin promotes melanin synthesis by enhancing the expression of key proteins involved in melanin synthesis, which may reduce senescence and apoptosis in B16F10 cells. Additionally, apigenin improves the imbalance between mitochondrial fission and fusion and delays the reduction in mitochondrial membrane potential to lessen ROS accumulation and mitigate mitochondrial damage.

Autophagy is a self-compromising measure taken by cells in the face of insufficient intracellular nutrition. It allows the cell to degrade and recycle certain intracellular components in an orderly manner to maintain the normal energy supply of the cell [32]. Lee KW et al. found that autophagy inhibitors or the knockdown of autophagy-related gene Atg5 could effectively restore the levels of pre-melanopsin and tyrosinase that had been reduced by the melanin synthesis inhibitors, and rapamycin, a non-selective autophagy inducer, could trigger α-MSH-stimulated cells to undergo autophagy, thereby promoting melanin synthesis [33]. We found that the expression of the autophagy-related protein was significantly inhibited when the cells underwent oxidative damage, and apigenin pretreatment promoted the expression of the above proteins in a dose-dependent manner (Figure 5H). This suggests that apigenin can ameliorate the oxidative damage of melanocytes and maintain the normal state and function of cells by promoting autophagy.

After mitochondrial damage, cells maintain the normal structure and function of mitochondria through reactions such as DNA repair and protein synthesis within mitochondria. In the study of mitochondrial autophagy, the PINK1/Parkin pathway has gained a lot of attention [34,35,36]; PINK1 is a protein kinase mainly found in mitochondrial membranes, and Parkin is an E3 ubiquitin ligase; they have synergistic effects and contribute to the activation of mitochondrial autophagy [37]. In addition, BNIP3L/Nix is a mitochondrial membrane surface-binding protein, and BNIP3L/Nix protein can directly connect with Atg8 family homologous proteins on the surface of phagocytosed membranes, thus inducing the degradation of damaged mitochondria via the autophagy pathway [38]. Mitochondrial autophagy is the last line of defense for the cell to deal with damaged mitochondria and ameliorate oxidative damage in cells [37,39]. BNIP3L/Nix is a class of outer mitochondrial membrane proteins that primarily operate as autophagosome recognition receptors during mitochondrial autophagy and begin mitochondrial autophagy by reacting with Parkin [38]. PINK1 and Parkin are Ser/Thr kinases and E3 ubiquitin ligases, respectively, which act synergistically to sense the functional state of the mitochondrion and function through the autophagy pathway to mark damaged mitochondria for autophagy [40]. We found that oxidative stress down-regulated the expression of the above proteins and inhibited mitochondrial autophagy, but apigenin ameliorated the inhibition of the expression of the above proteins, suggesting that apigenin ameliorates mitochondrial autophagy in melanocytes through activating the PINK1/Parkin pathway (Figure 6A–D).

The PI3K/Akt/mTOR signal pathway regulates autophagy. Upon activation of this pathway, cellular autophagy is inhibited [41]. We found that in the oxidative damage melanocytes, the PI3K/Akt/mTOR pathway was activated, and cellular autophagy was inhibited. The inhibition of the expression of the above proteins was found to be further enhanced by the combination of apigenin and the autophagy inhibitor in B16F10 cells (Figure 6E–K). This indicated that apigenin could activate the PINK1/Parkin pathway to improve mitochondrial autophagy and inhibit the PI3K/Akt/mTOR pathway to improve the autophagy of oxidatively damaged cells and delay apoptosis.

## 4. Materials and Methods

### 4.1. Materials

Apigenin, obtained from Sigma-Aldrich Corporation (CAS: 10789-100MG, with purity ≥ 95.0%, St. Louis, Missouri, MO, USA), was prepared with dimethyl sulfoxide (DMSO) at the concentration of 100 mM. It was stored as small aliquots at −20 °C. Before adding to cell culture medium, they were melted and diluted to appropriate concentration. Antibiotic penicillin/streptomycin, trypsin, and enhanced chemiluminescence (ECL) solution were purchased from Biosharp (Hefei, China). Dulbecco’s modified Eagle medium (DMEM) and fetal bovine serum (FBS) were purchased from GIBCO (Carlsbad, CA, USA). The cell lysis buffer and PI3K inhibitors (LY294002) were purchased from Biyotime (Nanjing, China). H_2_O_2_ (323381-25ML, with purity 3 wt.% in H_2_O, USA) was purchased from Sigma-Aldrich (St. Louis, MO, USA). Antibodies against GAPDH (60004, 1:1000), β-actin (3700, 1:2500), TYR (ab180753, 1:2000), TRP1 (ab3312, 1:200), TRP2 (ab74073, 1:1000), and gp100 (ab137078, 1:1000) were purchased from Abcam (Cambridge, UK). Keap1 (8047, 1:1000), Nrf2 (12721, 1:1000), HO-1 (43966, 1:1000), NQO1 (62262, 1:1000), Rac-1 (4651, 1:1000), Cdc42 (2466, 1:1000), E-Cadherin (3195, 1:1000), p62 (39749, 1:1000), p53 (2524, 1:1000), p21 (37543, 1:1000), MFF (84580, 1:1000), p-DRP1(S616) (4494, 1:1000), p-DRP1(S637) (6319, 1:1000), DRP1 (8570, 1:1000), Mitofusion-1 (14139, 1:1000), OPA1 (80471, 1:1000), Beclin-1 (2495, 1:1000), Atg5 (12994, 1:1000), Atg8 (64459, 1:1000), Atg12 (4180, 1:1000), LC3Ⅰ/II (12741, 1:1000), PI3K (4263, 1:1000), p-Akt (4060, 1:2000), Akt (9272, 1:1000), p-mTOR (5536, 1:1000), mTOR (2983, 1:1000), BNIP3L/Nix (12396, 1:1000), PINK1 (6946, 1:1000), Parkin (4211, 1:1000), second antibody anti-mouse IgG (7076, 1:2500), and anti-rabbit IgG (7074, 1:2500) were obtained from Cell Signaling Technology (Danvers, MA, USA). The BCA protein assay kit (P009), reactive oxygen species assay kit (DCFH-DA, S0033S), cellular senescence-associated β-galactosidase staining kit (C0602), and mitochondrial membrane potential assay kit (JC-1, CA1310) were purchased from Biyotime Biotechnology (Nanjing, China). The FITC–Phalloidin (No. 40735ES80) was purchased from Shanghai Yeasen Biotechnology Co. Ltd. (Shanghai, China). The Superoxide Dismutase (SOD) assay kit (E-BC-K020-M), Catalase (CAT) activity assay kit (E-BC-K031-S), and Malondialdehyde (MDA) assay kit (E-EL-0060) were purchased from Elabscience (Wuhan, China).

### 4.2. Cell Culture and Treatment

B16F10 cells were purchased from the Cell Bank (the Chinese Academy of Science, China, Shanghai). These cells were cultured in DMEM supplemented with 10% (*v*/*v*) FBS and 1% penicillin/streptomycin, and placed in a 37 °C, 5% CO_2_ incubator. An amount of 2.5% FBS was used when cells were treated with various concentrations of apigenin (0.01~10 μM) or under the condition of 48 h at 37 °C and with various concentrations of H_2_O_2_ (0.1~1 mM) to model oxidative stress for 12 h at 37 °C.

### 4.3. Zebrafish Feeding and Treatment

Adult zebrafish were obtained from Shanghai FishBio Co., Ltd. (China, Shanghai). Zebrafish were used to assess the depigmenting effects of pterostilbene in vivo with some modifications to a previous study [10]. Briefly, we kept the water temperature at about 28.5 °C, and they needed 14 h of light per day. Zebrafish embryos (collected after natural spawning was completed) were grouped into a 6-well plate by maintaining 30 embryos/well, and cultured in a light incubator at 28.5 °C. The embryos were treated or not with H_2_O_2_ (0.5 mM) for 4 h, in egg water containing apigenin (10 µM) for 24 h. The effects of apigenin on the ROS levels of zebrafish were observed using a Leica M205 FCA stereomicroscope (Germany, Heidelberg).

### 4.4. Determination of Cell Viability by the MTT Assay

The viability of B16F10 cells was determined by the MTT kit was purchased from Biyotime Biotechnology (China, Nanjing, C0009S-1). In brief, B16F10 cells were seeded in 96-well plates, 2–2.5 × 10^3^ cells per well, and placed in a 37 °C, 5% CO_2_ incubator for 24 h. After cell adherence, the FBS was decreased to 2.5%, and treated with different concentrations of apigenin (0~10 µM) and cultured for 48 h or treated with H_2_O_2_ (0~1 mM) and cultured for 12 h. MTT (0.5 mg/mL, 200 μL/well) treatment was applied, and samples were incubated at 37 °C for 4 h. Then, we removed the solution, dimethyl sulfoxide (DMSO) (150 μL/well) was added, and samples were oscillated gently for 5 min. The absorbance was detected at 570 nm [10].

### 4.5. FITC–Phalloidin Staining

In order to observe the cells’ morphological change, B16F10 cells were seeded in 10 cm cell culture dishes and treated with various concentrations of apigenin (0.1~5 µM) for 48 h, then treated with H_2_O_2_ (0.4 mM) for 12 h, and fixed with 3.7% paraformaldehyde for 10 min. Subsequently, refer to the staining instruction of FITC–Phalloidin was purchased from Shanghai Yeasen Biotechnology Co. Ltd. (Shanghai, China). Briefly, the fixed cell samples were stained with the FITC–Phalloidin fluorescence probe for 20~60 min, then washed with PBS 2~4 times, absorbing excess water. Then, the nuclei were stained with DAPI (blue fluorescence) and observed under the fluorescence microscope (Tokyo, Japan, Olympus) [10].

### 4.6. Intracellular Reactive Oxygen Species (ROS)

The effect of apigenin on ROS in B16F10 cells under oxidative stress or in zebrafish was detected by fluorescence probe DCFH-DA. Briefly, the cells were seeded on 6-well cell culture plates and treated with various concentrations of apigenin (0.1~5 µM) for 48 h, then treated with 0.4 mM H_2_O_2_ for 12 h. Zebrafish embryos were treated with egg water containing apigenin (10 µM) for 24 h, then cultured in 0.5 mM H_2_O_2_ for 4 h. Then, they were stained with the DCFH-DA fluorescence probe to visualize the intracellular ROS and observed by fluorescence microscope (Japan, Olympus). The fluorescence intensity was analyzed by ImageJ software version 1.53t [42].

### 4.7. Measurement of SOD Activity and CAT and MDA Levels

The Total Superoxide Dismutase (T-SOD, E-BC-K020-M) assay kit, Catalase (CAT) activity assay kit (E-BC-K031-S), and Malondialdehyde (MDA, E-EL-0060) assay kit were purchased from Elabscience (Wuhan, China). The SOD content and CAT and MDA activity were measured according to the instructions of each kit.

### 4.8. Determination of Mitochondrial Membrane Potential (MMP)

The alterations of MMP in B16F10 cells were analyzed by the JC-1 staining assay kit purchased from Biyotime Biotechnology (Nanjing, China, CA1310) according to the instructions. Briefly, the cells were seeded on 6-well culture plates and treated with various concentrations of apigenin (0.1~5 µM) for 48 h, then treated with H_2_O_2_ (0.4 mM) for 12 h; the cells were washed with PBS and then stained with JC-1 at 37 °C in the dark for 30 min. After being washed with staining buffer twice, they were observed by fluorescence microscope (Japan, Olympus) [43].

### 4.9. Senescence-Associated β-Galactosidase Assay

Senescence-associated β-galactosidase activity in senescent cells was elevated by a senescence cells histochemical staining kit purchased from Biyotime Biotechnology (Nanjing, China, C0602). Briefly, cells were seeded on 6-well culture plates and treated with various concentrations of apigenin (0.1~5 µM) for 48 h, then cultured with 0.4 mM H_2_O_2_ for 12 h. Finally, cells were stained according to the manufacturer’s instructions. Senescence-associated β-galactosidase was stained blue. We used a light microscope (400×) to collect images [29].

### 4.10. Western Blot Analysis

Total cell protein was obtained according to the above-mentioned method after the determination of the protein concentration by the BCA assay kit purchased from Biyotime Biotechnology (Nanjing, China, P009). Then, the lysates were denatured by heating and boiling at 100 °C for 10 min for Western blot assay. The proteins were separated on 8%, 10%, 12%, and 15% SDS-PAGE gels and then transferred into the polyvinylidene fluoride membranes (PVDF membranes). Subsequently, the PVDF membranes were blocked with 5% skimmed milk at 24 °C for 2 h, washed three times with TBST, then incubated with primary antibodies overnight at 4 °C, and then incubated with secondary antibodies for 2 h at 24 °C. Finally, the PVDF membranes were visualized using an ECL reagent, and detected with a luminous imaging system (Tanon-4160). The results shown here are representative of at least three repeated experiments [10].

### 4.11. Statistical Analysis

All the experiments were repeated three times, and all data are presented as the mean ± standard error of the mean (Mean ± SEM). All were processed by GraphPad prism 9, and statistical analysis was performed with one-way ANOVA (and nonparametric) followed by Turkey’s post hoc test for multiple comparisons. Values of *p* < 0.05 indicated statistical significance.

## 5. Conclusions

In summary, this study proved that apigenin is a potent anti-melanogenic agent and has derived the following conclusions: (1) Apigenin exerts its antioxidant effects and delays cellular senescence and apoptosis by activating the Nrf2 pathway, inhibiting the p53/p21 pathway, and facilitating the fusion of mitochondria to delay the mitochondrial membrane potential decrease. (2) Apigenin could inhibit the PI3K/Akt/mTOR pathway, and activate the PINK1/Parkin signal pathway, to ameliorate cellular oxidative damage. Our study revealed the potential applicability of apigenin in the treatment of hyperpigmentation disorders and vitiligo.

## Figures and Tables

**Figure 1 pharmaceuticals-17-01302-f001:**
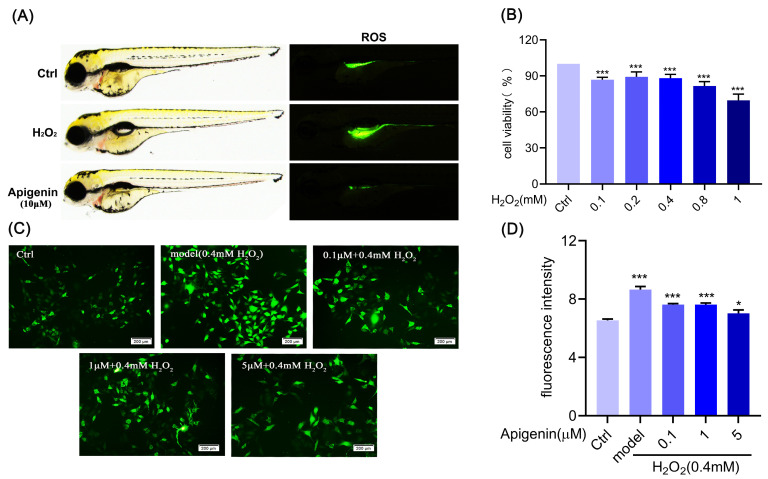
Apigenin repressed oxidative stress-induced ROS production. (**A**) Effects of apigenin (10 µM) on ROS level of zebrafish. Zebrafish were treated with egg water containing apigenin (10 µM) for 24 h, then treated with H_2_O_2_ (0.5 mM) for 4 h. DCFH-DA (10 µM) was added to zebrafish for 30 min. Images were captured by fluorescence microscope. (**B**) Treated with various concentrations (0.1~1 µM) of H_2_O_2_ for 12 h; relative cell viability was determined by MTT assay (n = 7). (**C**,**D**) B16F10 cells were pretreated with apigenin (0.1, 1, and 5 µM) for 48 h, then treated with H_2_O_2_ (0.4 mM) for 12 h. Intracellular ROS levels were indicated by DCFH-DA fluorescence probe. Intracellular ROS changes in B16F10 cells were observed by 200× fluorescence microscope, Bar = 200 μm. Data are expressed as mean ± SEM (n = 3). * *p* < 0.05, and *** *p* < 0.001 vs. control.

**Figure 2 pharmaceuticals-17-01302-f002:**
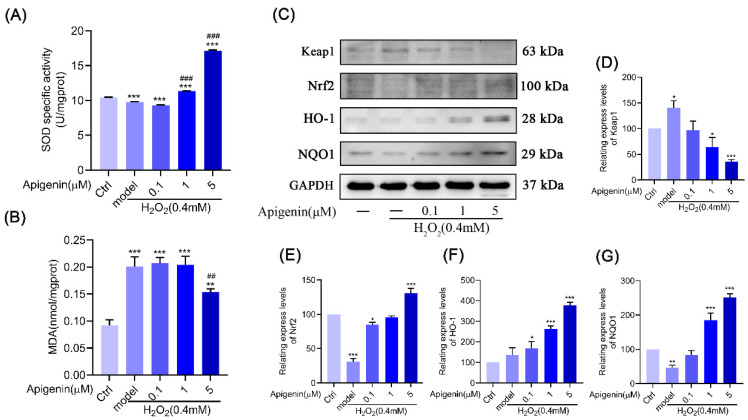
Apigenin activated Nrf2 pathway. (**A**,**B**) Measurement of SOD activity and MDA contents in B16F10 cells. Data are presented as the mean ± SEM (n = 3). (**C**–**G**) Effects of apigenin (0.1, 1, and 5 µM) on Nrf2 pathway in oxidative stress B16F10 cells. Treated with apigenin (0.1, 1, and 5 µM) for 48 h and then treated with H_2_O_2_ (0.4 mM) for 12 h in B16F10 cells, and WB was then applied to detect the protein levels of Keap1, Nrf2, HO-1, and NQO1. Data are expressed as the mean ± SEM (n = 3). * *p* < 0.05, ** *p* < 0.01, and *** *p* < 0.001 vs. control. ^##^ *p* < 0.01 and ^###^ *p* < 0.001, vs. model.

**Figure 3 pharmaceuticals-17-01302-f003:**
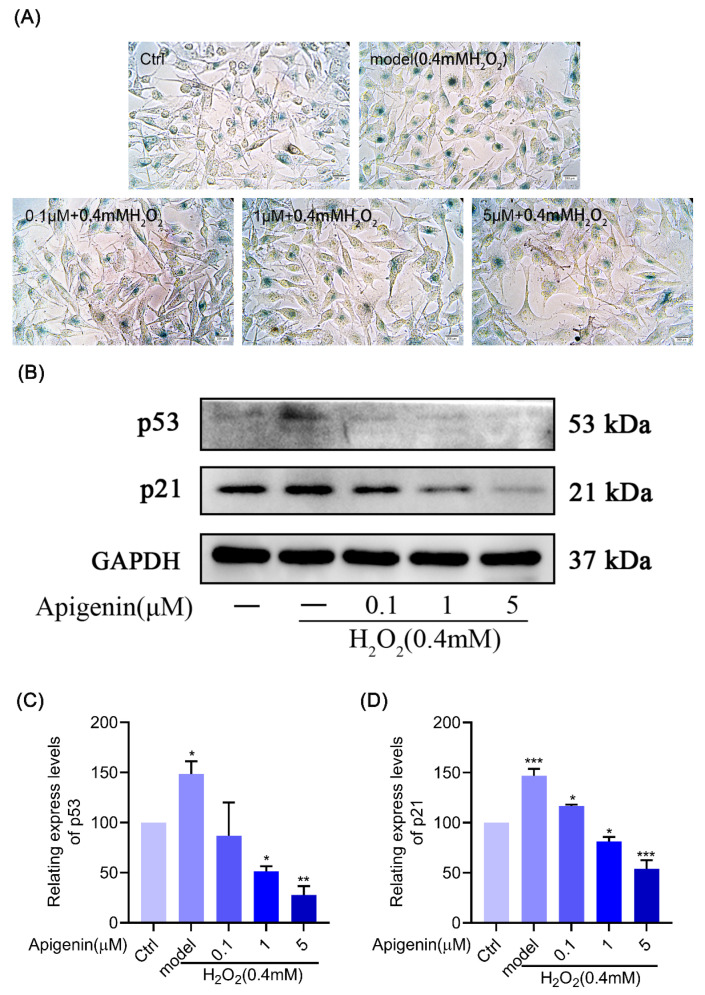
Apigenin inhibits cellular senescence induced by oxidative stress in B16F10 cells. (**A**) Effect of apigenin on senescence-associated β-galactosidase of B16F10 cells. Cells were seeded on 6-well culture plates and treated with various concentrations of apigenin (0.1, 1, and 5 µM) for 48 h, then treated with H_2_O_2_ (0.4 mM) for 12 h. Senescence-associated β-galactosidase was stained blue. Intracellular color changes in B16F10 cells were observed by microscope at 400× magnification. (**B**–**D**) Apigenin inhibits the expression of p53 and p21 in B16F10 cells. Treated with apigenin (0.1, 1, and 5 µM) for 48 h and then treated with H_2_O_2_ (0.4 mM) for 12 h in B16F10 cells, and WB was then applied to detect the protein levels of p53 and p21. Data are expressed as the mean ± SEM (n = 3). * *p* < 0.05, ** *p* < 0.01, and *** *p* < 0.001 vs. control.

**Figure 4 pharmaceuticals-17-01302-f004:**
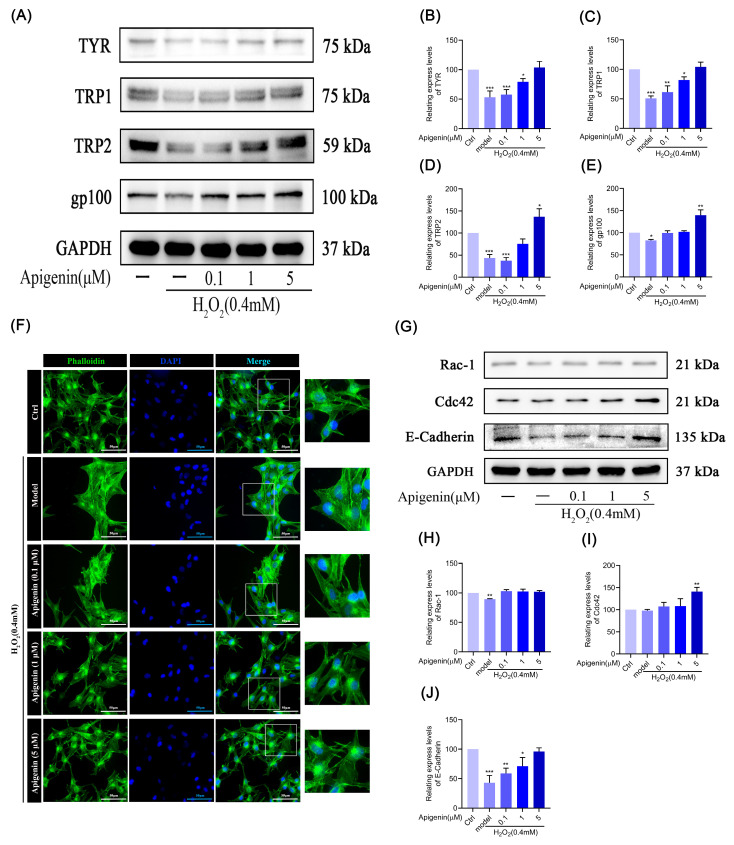
Apigenin ameliorates oxidative stress-induced impairment of melanin synthesis and dendritic atrophy. (**A**–**E**) B16F10 cells were treated with apigenin (0.1, 1, and 5 µM) for 48 h and then treated with H_2_O_2_ (0.4 mM) for 12 h, and WB was then applied to detect the protein levels of TYR, TRP1, TRP2, and gp100. (**F**) Treated with apigenin (0.1, 1, and 5 µM) for 48 h and then treated with H_2_O_2_ (0.4 mM) for 12 h in B16F10 cells, stained with FITC–Phalloidin fluorescence probe to visualize the cytoskeleton. (**G**–**J**) B16F10 cells were treated with apigenin (0.1, 1, and 5 µM) for 48 h and then treated with H_2_O_2_ (0.4 mM) for 12 h, and WB was then applied to detect the protein levels of Rac-1, Cdc42, and E-Cadherin. Data are expressed as the mean ± SEM (n = 3). * *p* < 0.05, ** *p* < 0.01, and *** *p* < 0.001 vs. control.

**Figure 5 pharmaceuticals-17-01302-f005:**
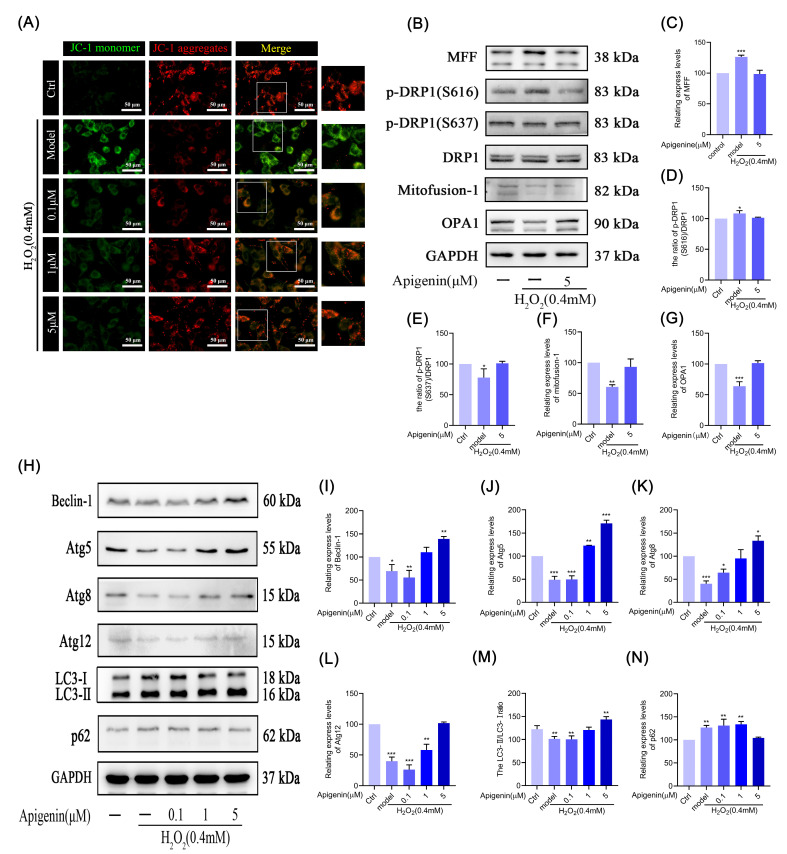
Apigenin ameliorates oxidative stress-induced mitochondrial damage and cellular autophagy inhibition. (**A**) Effect of apigenin on mitochondrial membrane potential in oxidant model of B16F10 cells. B16F10 cells were treated with apigenin (0.1, 1, and 5 µM) for 48 h, then cultured in H_2_O_2_ (0.4 mM) for 12 h. Mitochondrial membrane potential was stained with JC-1. Intracellular color changes in oxidant model of B16F10 cells were observed by fluorescence microscope at 400× magnification. (**B**–**N**) B16F10 cells were treated with apigenin (0.1, 1, and 5 µM) for 48 h and then treated with H_2_O_2_ (0.4 mM) for 12 h, and WB was then applied to detect protein levels of MFF, p-DRP1(S616), p-DRP1(S637), DRP1, Mitofusion1, OPA1, Beclin-1, Atg5, Atg8, Atg12, LC3-Ⅰ, LC3-II, and p62. Data are expressed as mean ± SEM (n = 3). * *p* < 0.05, ** *p* < 0.01, and *** *p* < 0.001 vs. control.

**Figure 6 pharmaceuticals-17-01302-f006:**
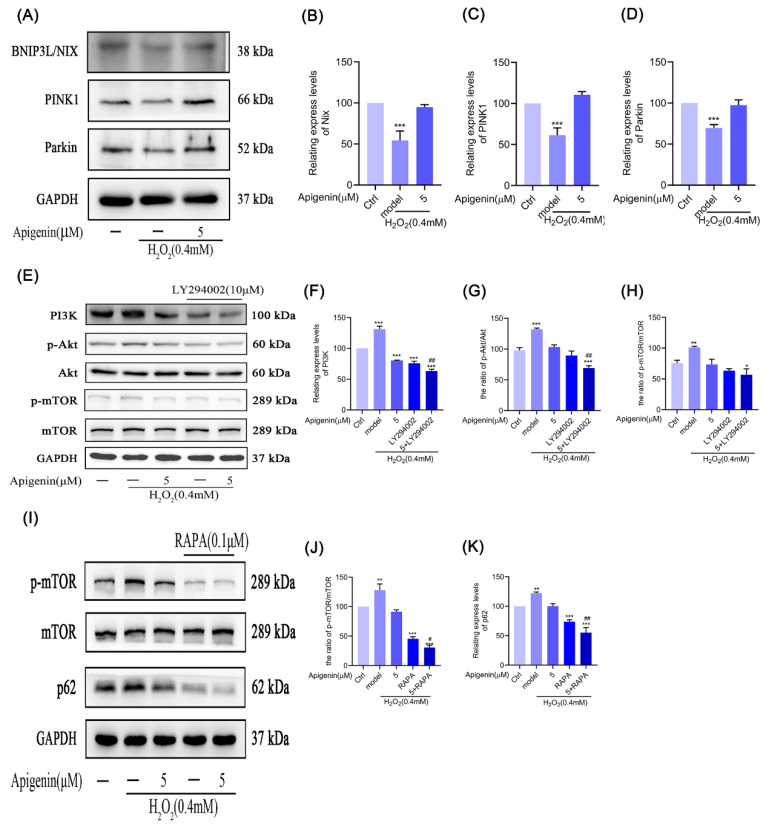
The effects of apigenin on the expression of BNIP3L/Nix, PINK1, and Parkin and the activity of the PI3K/Akt/mTOR signaling pathway in melanocytes. (**A**–**D**) The B16F10 cells were treated with apigenin (5 µM) for 48 h and then treated with H_2_O_2_ (0.4 mM) for 12 h, and WB was then applied to detect the protein levels of BNIP3L/Nix, PINK1, and Parkin. (**E**–**K**) The B16F10 cells were pretreated with LY294002 (10 µM) or RAPA (0.1 µM) for 2 h and incubated with apigenin (5 μM) for 24 h, then cells were treated with H_2_O_2_ (0.4 mM) for 12 h. The protein levels of PI3K, p-Akt, Akt, p-mTOR, mTOR, and p62 were determined by Western blot. Data are expressed as the mean ± SEM (n = 3). * *p* < 0.05, ** *p* < 0.01, and *** *p* < 0.001 vs. control. ^#^
*p* < 0.05 and ^##^
*p* < 0.01, (**B**): 5+LY294002 vs. LY294002; (**C**): 5+RAPA vs. RAPA.

## Data Availability

The original contributions presented in the study are included in the article/Appendix A, further inquiries can be directed to the corresponding authors.

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
