# Peer review of "Apigenin Ameliorates H2O2-Induced Oxidative Damage in Melanocytes through Nuclear Factor-E2-Related Factor 2 (Nrf2) and Phosphatidylinositol 3-Kinase (PI3K)/Protein Kinase B (Akt)/Mammalian Target of Rapamycin (mTOR) Pathways and Reducing the Generation of Reactive Oxygen Species (ROS) in Zebrafish"

_pharmaceuticals, 2024, doi:10.3390/ph17101302_

Round 1

Reviewer 1 Report

Comments and Suggestions for Authors

Dear respected Editor of Pharmaceuticals,

Many thanks for your confidence for peer reviewing of the original research article entitled “Apigenin Ameliorates H2O2-Induced Oxidative Damage in Melanocytes through Nrf2 and PI3K/Akt/mTOR Pathways for consideration in Pharmaceuticals with Manuscript ID: pharmaceuticals-3205941

The study presents important data. Yet, several concerns exist and wide revisions are required to be fit for publication:

Major concerns

1.     More reviewing data about the flavonoid apigenin should be presented in the section of introduction

2.     Zebrafish Feeding and Treatment lines 382-390: more illustration is needed as authors mentioned that this experiment was done to assess the depigmenting effects of pterostilbene in vivo. Mention how it was assessed in relation to use of H2O2, apigenin, ROS. Additionally, the different concentrations used in the study assays as for such assay and MTT assays the concentration of apigenin was (10 μL) while in the FITC staining was (0.5-1 μL) please illustrate.

Other concerns

1.     Title refers only to in vitro study on melanocytes but doesn’t illustrate the study on zebra fish embryos, please rewrite again taking in consideration both in vivo and in vitro studies

2.     The abstract needs to be reorganized to clarify the experimental protocols first then the results. Also, the background needs to be shortened. Abstract obscures the important results and findings. Please modify it accordingly.

3.     Line 18-20 what is the dose level used, duration, cell lines used

4.     Line 16, 36: grammatical errors, has minor grammatical and structural errors. Please, double-check. English could be improved.

5.     The full names should be provided when abbreviations appear for the first time. Please ensure that all abbreviations in the manuscript are introduced with their full terms upon first mention. Line 19: MTT, line 24: ROS, MDA, line 47: PMEL, line 378: DMEM, line 399: FITC, line 422: JC-1, line 437: BCA,

6.     Introduction line 67-71: authors should present the main aim of the study or what the problem the research is directed to resolve. Thus, the obtained results of the current research should not be mentioned in the section of introduction, focusing only on your study aim

7.     Citing references for paragraph at lines 62-65, lines 40-42 should be mentioned, lines 278-279

8.     Lines 295-299: authors mentioned the obtained results which is not fully discussed please rewrite accordingly.

9.     Line 371-375: CAT. Number for the used kits please supply

10.  Line 380: mention the doses or selected apigenin concentrations

11.  Line 420 mention kits catalog number, also for kits used in lines 430 and 437

12.  The immune-histochemical technique should be illustrated in more detail particularly the primary and secondary antibody used.

13.  References to most protocols used are missed like

14.  In all figure legends and table footnotes: the data has been presented as means± SEM but the number of replicates n=? Also, the full term of all abbreviations used should be clarified

15.  Figure 1 is overcrowded that make it difficult to be clarified please modify

Comments on the Quality of English Language

Dear respected Editor of Pharmaceuticals,

Many thanks for your confidence for peer reviewing of the original research article entitled “Apigenin Ameliorates H2O2-Induced Oxidative Damage in Melanocytes through Nrf2 and PI3K/Akt/mTOR Pathways for consideration in Pharmaceuticals with Manuscript ID: pharmaceuticals-3205941

The study presents important data. Yet, several concerns exist and wide revisions are required to be fit for publication:

Major concerns

1.     More reviewing data about the flavonoid apigenin should be presented in the section of introduction

2.     Zebrafish Feeding and Treatment lines 382-390: more illustration is needed as authors mentioned that this experiment was done to assess the depigmenting effects of pterostilbene in vivo. Mention how it was assessed in relation to use of H2O2, apigenin, ROS. Additionally, the different concentrations used in the study assays as for such assay and MTT assays the concentration of apigenin was (10 μL) while in the FITC staining was (0.5-1 μL) please illustrate.

Other concerns

1.     Title refers only to in vitro study on melanocytes but doesn’t illustrate the study on zebra fish embryos, please rewrite again taking in consideration both in vivo and in vitro studies

2.     The abstract needs to be reorganized to clarify the experimental protocols first then the results. Also, the background needs to be shortened. Abstract obscures the important results and findings. Please modify it accordingly.

3.     Line 18-20 what is the dose level used, duration, cell lines used

4.     Line 16, 36: grammatical errors, has minor grammatical and structural errors. Please, double-check. English could be improved.

5.     The full names should be provided when abbreviations appear for the first time. Please ensure that all abbreviations in the manuscript are introduced with their full terms upon first mention. Line 19: MTT, line 24: ROS, MDA, line 47: PMEL, line 378: DMEM, line 399: FITC, line 422: JC-1, line 437: BCA,

6.     Introduction line 67-71: authors should present the main aim of the study or what the problem the research is directed to resolve. Thus, the obtained results of the current research should not be mentioned in the section of introduction, focusing only on your study aim

7.     Citing references for paragraph at lines 62-65, lines 40-42 should be mentioned, lines 278-279

8.     Lines 295-299: authors mentioned the obtained results which is not fully discussed please rewrite accordingly.

9.     Line 371-375: CAT. Number for the used kits please supply

10.  Line 380: mention the doses or selected apigenin concentrations

11.  Line 420 mention kits catalog number, also for kits used in lines 430 and 437

12.  The immune-histochemical technique should be illustrated in more detail particularly the primary and secondary antibody used.

13.  References to most protocols used are missed like

14.  In all figure legends and table footnotes: the data has been presented as means± SEM but the number of replicates n=? Also, the full term of all abbreviations used should be clarified

15.  Figure 1 is overcrowded that make it difficult to be clarified please modify

Sincerely,

Ehsan

Author Response

For research article

Response to Reviewer 1 Comments

1. Summary

2. Questions for General Evaluation

Reviewer’s Evaluation

Response and Revisions

Does the introduction provide sufficient background and include all relevant references?

Can be improved

3. Point-by-point response to Comments and Suggestions for Authors

Major concerns

Comments 1: More reviewing data about the flavonoid apigenin should be presented in the section of introduction

Response 1: Thank you for pointing this out. We agree with this comment. Therefore, we have added a more synoptic description of the flavonoid apigenin in the section of introduction, on page 5, paragraph 1, lines 80-89 of the revised manuscript.

Comments 2: Zebrafish Feeding and Treatment lines 382-390: more illustration is needed as authors mentioned that this experiment was done to assess the depigmenting effects of pterostilbene in vivo. Mention how it was assessed in relation to use of H2O2, apigenin, ROS. Additionally, the different concentrations used in the study assays as for such assay and MTT assays the concentration of apigenin was (10 μL) while in the FITC staining was (0.5-1 μL) please illustrate.

Response 2: Thank you very much for your careful review, in lines 382-390 we mentioned the decolourising effect of pterostilbene in zebrafish, so we used the method in this article to assess the effect of apigenin on the pigmentation in zebrafish, the results of which we placed in the Supplementary Material in Fig. S1. The results of the MTT assay to detect the effect of apigenin on the viability of B16F10 cells are also placed in the Supplementary Material in Fig. S2.

Other concerns

1.  Title refers only to in vitro study on melanocytes but doesn’t illustrate the study on zebra fish embryos, please rewrite again taking in consideration both in vivo and in vitro studies

Response 1: Thank you for pointing this out. We agree with this observation. Therefore, we have revised the article title to Apigenin ameliorates H2O2-induced oxidative damage through Nrf2 and PI3K/Akt/mTOR pathways. Thank you again for your suggestion, and we hope that you will read the revised article title, and if it is not appropriate, we hope that you will make your suggestion and we will revise it.

2.  The abstract needs to be reorganized to clarify the experimental protocols first then the results. Also, the background needs to be shortened. Abstract obscures the important results and findings. Please modify it accordingly.

Response 2: Thank you very much for your suggestion, we have changed the summary section accordingly. A detailed description of the results section of the abstract has also been provided. It is on pages 2-3, lines 16-55 of the manuscript. Thank you again for your time and effort on this article, please let us know if there is any inappropriateness and need for modification and we will make changes according to your suggestions.

3.  Line 18-20 what is the dose level used, duration, cell lines used

Response 3: Thank you very much for your careful review, we have revised the methodology in the abstract, It is on page 2, lines 22-23 of the manuscript, if there is still something wrong after the revision, I hope you can give us suggestions again, we will be very grateful to you and make changes according to your suggestions.

4.  Line 16, 36: grammatical errors, has minor grammatical and structural errors. Please, double-check. English could be improved.

Response 4: Thank you very much for pointing out our grammatical errors, and we apologise for any reading inconvenience caused by our carelessness. We have made the appropriate changes. Thank you again for your seriousness!

5.  The full names should be provided when abbreviations appear for the first time. Please ensure that all abbreviations in the manuscript are introduced with their full terms upon first mention. Line 19: MTT, line 24: ROS, MDA, line 47: PMEL, line 378: DMEM, line 399: FITC, line 422: JC-1, line 437: BCA,

Response 5: Thank you very much for your suggestion. The full names of the corresponding abbreviations are already listed in the ‘Abbreviations’ section of the manuscript, but probably because the file generated after submission does not show the contents of the abbreviations, we have now included the abbreviations as follows.

Abbreviations

Akt, protein kinase (PKB); BNIP3L/Nix, Bcl-2/adenovirus E1B 19 kDa protein-interacting protein 3-like; Cdc42, Cell division cycle 42; DCFH-DA, 2,7-Dichlorod -hydrofluorescein diacetate; DRP1, dynamin-related protein 1; FBS, Fetal bovine serum; GAPDH, Glyceraldehy phosphate dehydrogenase; HO-1, Heme Oxygenase-1; Keap1, Recombinant Kelch Like ECH Associated Protein 1; MFF, Mitochondrial Fission Factor; MDA, Malondialdehyde; MTT, Methylthiazolyl tertrazolium bormide; MMP, Mitochondrial membrane potential; mTOR, mammalian target of Rapamycin; NQO1, NAD(P)H: quinone oxidoreductase 1; Nrf2, Nuclear factor-E2-related factor 2; PAGE, Polyacrylamide gel electrophoresis; PVDF, Polyvinylidene Fluoride; PBS, Phosphate buffered saline; PMSF, Phenylmethl sulfonylfluoride; PI3K, phosphatidylinositol 3-kinase; Rac-1, Ras-related C3 botulinum toxin substrate 1; ROS, Reactive oxygen species; SDS, Sodium didecylsulfate; SOD, Superoxide dismutase; TRP1, tyrosinase-related protein 1; TRP2, dopachrome tautomerase; TYR, Tyrosinase.

6.  Introduction line 67-71: authors should present the main aim of the study or what the problem the research is directed to resolve. Thus, the obtained results of the current research should not be mentioned in the section of introduction, focusing only on your study aim

Response 6: Thank you very much for your suggestion, and we apologise for the inconvenience caused by our carelessness, and have now removed the reference to the current findings section from the introduction.

7.  Citing references for paragraph at lines 62-65, lines 40-42 should be mentioned, lines 278-279

Response 7: Thank you very much for your suggestion, and we apologise for the inconvenience caused by our carelessness, and have now revised the corresponding section.

8.  Lines 295-299: authors mentioned the obtained results which is not fully discussed please rewrite accordingly.

Response 8: Thank you very much for pointing out our shortcomings, and we apologise for the inconvenience our carelessness has caused you in reading. We have already revised the content of the section according to your suggestions and made detailed changes, if there is still a need for changes, please give us suggestions again and we will try our best to make changes.

9.  Line 371-375: CAT. Number for the used kits please supply

Response 9: Thank you very much for your careful attention. We have written the CAT. Number on the back of the corresponding kits.

10.  Line 380: mention the doses or selected apigenin concentrations

Response 10: Thank you very much for your careful attention. We have written the kit numbers on the back of the corresponding kits, which are highlighted in red in the revised manuscript.

11.  Line 420 mention kits catalog number, also for kits used in lines 430 and 437

Response 11: Thank you very much for your careful attention. We have written the kit numbers on the back of the corresponding kits, which are highlighted in red in the revised manuscript.

12.  The immune-histochemical technique should be illustrated in more detail particularly the primary and secondary antibody used.

Response 12: Thank you for your suggestion, the primary antibody and secondary antibody used in WB have been listed with corresponding item numbers and dilution ratios, if there is still a need for improvement, please point out the parts that need to be improved in detail, and we will make modifications according to your suggestions.

13.  References to most protocols used are missed like

Response 13: Thank you for your suggestions, we have some not quite understand what you mean, I hope you can tell us in detail the details of the need to change, we will follow your suggestions for further modification and improvement.

14.  In all figure legends and table footnotes: the data has been presented as means± SEM but the number of replicates n=? Also, the full term of all abbreviations used should be clarified

Response 14: Thank you very much for your careful review, the data in this manuscript are expressed as mean ± SEM, number of replicates n=3.

15.  Figure 1 is overcrowded that make it difficult to be clarified please modify

Response 15: Thank you for your suggestions, we have some not quite understand what you mean, I hope you can tell us in detail the details of the need to change, we will follow your suggestions for further modification and improvement.

Once again, we sincerely thank you for your suggestions on this manuscript, and if there are still deficiencies after the revision, we hope you can give us your valuable suggestions, we will be grateful!

Reviewer 2 Report

Comments and Suggestions for Authors

a)     The authors should check for typographical and grammar error the entire manuscript (space, page lines etc…)

b)    Authors, please add references for oxidative stress and his role in the development of vitiligo.

c)     Authors, please explain better the aim of the study and possible future studies.

d)    Add references for Nrf2 and the pathogenesis of vitiligo.

e)     western blots, although repeated, do not show clear results of protein expression in figures 1 and 2. The authors are requested to provide clearer results.

f)     the expression of SOD has been investigated; why not the expression of CAT?

g)    Authors are requested to add more information about Assay kit.

h)    Pathways such as F-actin, Cdc42, Rac-1 and E-Cadherin are investigated which are not mentioned in the introduction. Please argue the connection between these and the purpose of the work.

i)      The fluorescence shown in figure 3 is over-expressed. There are no differences between the various experimental groups; please change them.

j)      What is the limitation of the study?

k)    The western blots in figure 5 show a different pattern from the others. what is the explanation? the western blots must all have the same expression. (- - 0.1 1 5 as in the previous ones).

l)      Nrf2, in western blot, is expressed from total cell lysate; the authors are requested to highlight the nuclear pattern and normalise it to laminin, the cellular pattern to GAPDH.

Author Response

For research article

Response to Reviewer 2 Comments

1. Summary

2. Questions for General Evaluation

Reviewer’s Evaluation

Response and Revisions

Does the introduction provide sufficient background and include all relevant references?

Must be improved

Thank you for your careful review, and we have revised the manuscript accordingly, following your suggestions.

Are all the cited references relevant to the research?

Must be improved

Thank you for your careful review, and we have revised the manuscript accordingly, following your suggestions.

Is the research design appropriate?

Can be improved

Thank you for your careful review, and we have revised the manuscript accordingly, following your suggestions.

Are the methods adequately described?

Must be improved

Thank you for your careful review, and we have revised the manuscript accordingly, following your suggestions.

Are the results clearly presented?

Must be improved

Thank you for your careful review, and we have revised the manuscript accordingly, following your suggestions.

Are the conclusions supported by the results?

Can be improved

Thank you for your careful review, and we have revised the manuscript accordingly, following your suggestions.

3. Point-by-point response to Comments and Suggestions for Authors

Comments 1: The authors should check for typographical and grammar error the entire manuscript (space, page lines etc…)

Response 1: Thank you for your careful review and for pointing out our mistakes, we are very sorry for the inconvenience caused by our carelessness. We have carefully checked the manuscript layout and the spaces again according to your suggestions, if there is still something wrong, please let us know and we will definitely revise and improve according to your suggestions.

Comments 2: Authors, please add references for oxidative stress and his role in the development of vitiligo.

Response 2: Thank you very much for your suggestion, we have added references for oxidative stress and his role in the development of vitiligo. On page 6 of the manuscript, second paragraph, lines 109-127.

Comments 3: Authors, please explain better the aim of the study and possible future studies.

Response 3: Thank you for your question. This manuscript explores the ameliorative effect of apigenin on premature melanocyte aging and loss and its mechanism. We first examined the effects of apigenin on melanin synthesis and transport in melanocytes under oxidative stress states and the mechanisms. Second, we disrupted the redox balance of melanocytes by hydrogen peroxide to establish a model of oxidative stress similar to that of premature melanocyte aging and loss in the course of vitiligo. Using this model, we explored the effects and mechanisms of apigenin on premature melanocyte aging and loss. Finally, we also investigated the effect of apigenin on the level of mitochondrial autophagy in melanocytes under oxidative stress. Through these studies, we hope to provide a theoretical basis for the further development of apigenin in the treatment of vitiligo.

Comments 4: Add references for Nrf2 and the pathogenesis of vitiligo.

Response 4: Thank you very much for your suggestion, we have added references for Nrf2 and the pathogenesis of vitiligo. On pages 6-7 of the manuscript, second paragraph, lines 128-143 .

Comments 5: western blots, although repeated, do not show clear results of protein expression in figures 1 and 2. The authors are requested to provide clearer results.

Response 5: Thank you for your suggestion, this is the clearest band of the duplicate experiments we have used. We apologise for any inconvenience this may have caused you in reading.

Comments 6: the expression of SOD has been investigated; why not the expression of CAT?

Response 6: Thank you very much for your careful review, and based on your suggestions, we have supplemented the results of CAT's expression by placing them in Figure S3 in the Supplementary Material.

Comments 7: Authors are requested to add more information about Assay kit.

Response 7: Thank you very much for your careful attention. We have written the CAT. Number on the back of the corresponding kits.

Comments 8: Pathways such as F-actin, Cdc42, Rac-1 and E-Cadherin are investigated which are not mentioned in the introduction. Please argue the connection between these and the purpose of the work.

Response 8: Rac1 mediates the formation and elongation of plate pseudopods, and Cdc42 mediates the formation and elongation of filamentous pseudopods. E-cadherin is the main mediator of adhesion between normal human epidermal melanocytes and keratinocytes, and the expression of the above proteins is positively correlated with melanin transport. Therefore, we detected the expression of the above proteins by Western blot in order to investigate the effect of apigenin on the oxidative stress state of the effect of apigenin on the transport of B16F10 cells under oxidative stress.

Comments 9: The fluorescence shown in figure 3 is over-expressed. There are no differences between the various experimental groups; please change them.

Response 9: Thank you for your suggestion, we have changed the image corresponding to Figure 3. If there is still something wrong, please point it out to us and we will correct it carefully!

Comments 10: What is the limitation of the study?

Response10: Thank you for your question. The mechanism of apigenin's melanocyte-promoting effect and antioxidant pharmacological activity needs to be further demonstrated by inhibitors or gene knockdown. The promotional effect of apigenin on mitochondrial autophagy and fusion under oxidative stress has only been preliminarily explored and needs further research. The pharmacological activity of apigenin in this paper was mainly demonstrated in cells, which needs to be further verified in animal experiments.

Comments 11: The western blots in figure 5 show a different pattern from the others. what is the explanation? the western blots must all have the same expression. (- - 0.1 1 5 as in the previous ones).

Response 11: The PI3K/Akt/mTOR signaling pathway is a pathway that regulates autophagy. Upon activation of this pathway, cellular autophagy is inhibited doi: 10.1007/s00253-019-10257-8. The downstream signaling molecule mTOR, a serine/threonine kinase, is a major regulator of cellular metabolism doi: 10.3389/fcell.2021.655731. mTOR activation can inactivate a number of autophagy-associated proteins by phosphorylating them, leading to the inhibition of autophagy, e.g., ULK1, which is a protein that promotes the initiation of autophagy and autophagic niche formation. pathway to promote autophagy in melanocytes under oxidative stress. We found that the expression of PI3K and phosphorylation of Akt and mTOR, key proteins in this pathway, were up-regulated in melanocytes under oxidative stress, and the PI3K/Akt/mTOR signaling pathway was activated.

was activated and cellular autophagy was inhibited. In contrast, both apigenin (5 µM) and the PI3K protein inhibitor LY294002 resulted in the oxidative stress-induced up-regulation of PI3K expression and phosphorylation of Akt and mTOR, and the activation of this pathway was inhibited. We used the PI3K inhibitor LY294002 (inhibition of the PI3K/Akt/mTOR signaling pathway)

Co-treatment with apigenin revealed a synergistic effect with further enhancement of inhibition relative to apigenin or LY294002 alone. We also co-treated apigenin with rapamycin RAPA, an inhibitor of mTOR phosphorylation downstream of the PI3K/Akt/mTOR signaling pathway, and found that its inhibitory effect on mTOR phosphorylation was greater than that of apigenin or LY294002 alone, resulting in a synergistic effect.

The inhibitory effect on mTOR phosphorylation was found to be further enhanced relative to apigenin or RAPA alone, resulting in a synergistic effect and further enhancement of autophagy. This demonstrates that apigenin can promote autophagy in melanocytes under oxidative stress by inhibiting the PI3K/Akt/mTOR signaling pathway.

Comments 12: Nrf2, in western blot, is expressed from total cell lysate; the authors are requested to highlight the nuclear pattern and normalise it to laminin, the cellular pattern to GAPDH.

Response 12: Thank you for your careful review and suggestions, we have followed your suggestion and detected the expression of Nrf2 in the nucleus by Western blot and placed the results in the Supplementary Material Figure S5, and added expression of Nrf2 in the cytosol in the Supplementary Material Figure S4.

Once again, we sincerely thank you for your suggestions on this manuscript, and if there are still deficiencies after the revision, we hope you can give us your valuable suggestions, we will be grateful!

Reviewer 3 Report

Comments and Suggestions for Authors

In this study authors investigated the effects of apigenin on the mechanism of repairing oxidative cell damage and found that apigenin significantly promoted melanogenesis in B16F10 cells and zebrafish embryos. Moreover, apigenin activated the Nrf2 pathway increasing SOD expression and reducing ROS. Apigenin also inhibited the expression of senescence-related proteins p53/p21, promoted mitochondrial fusion, and delayed the decrease of mitochondrial membrane potentia. Apigenin reduced oxidative damage by inhibiting the PI3K/Akt/mTOR pathway and activating the PINK1/Parkin pathway. 

The manuscript and topic are very interesting. However, several points deserve to be improved. In particular:

Introduction: Although NRF2/KEAP1 signaling plays a key role in this manuscript, this signaling is not even mentioned in the introduction. However, the multifaceted role of this important pathway deserves to be highlithed since it plays a key role in several pathology including cancer (see PMID: 37296999 ). This is an important point to state since it can further highlight the interesting results obtained by the authors. 

Lines 40-42; 61-65: references are needed

Line 99: ARE regions are located in the promoters of genes coding for antioxidant enzymes

Figure 1C: Images are too small

Figure 2A: scale bars are needed

Figure 3F and 4A: images are too small to appreciate the staining localization

Why authors did not use normal human melanocyte? 

4.1. Materials: I suggest to move the antibodies used in a dedicate table 

Authors must report the product code of all kits used

Abbreviations must be written in full length when mentioned for the first time

An accurate revision of syntax is recommended

Comments on the Quality of English Language

An accurate revision of syntax is recommended

Author Response

For research article

Response to Reviewer 3 Comments

1. Summary

2. Questions for General Evaluation

Reviewer’s Evaluation

Response and Revisions

Does the introduction provide sufficient background and include all relevant references?

Must be improved

Are all the cited references relevant to the research?

Must be improved

Is the research design appropriate?

Can be improved

Are the methods adequately described?

Can be improved

Are the results clearly presented?

Must be improved

Are the conclusions supported by the results?

Can be improved

3. Point-by-point response to Comments and Suggestions for Authors

Comments 1: Introduction: Although NRF2/KEAP1 signaling plays a key role in this manuscript, this signaling is not even mentioned in the introduction. However, the multifaceted role of this important pathway deserves to be highlithed since it plays a key role in several pathology including cancer (see PMID: 37296999IF: 4.5 Q1  ). This is an important point to state since it can further highlight the interesting results obtained by the authors.

Response 1: Thank you very much for your suggestion, we have added references for Nrf2 and the pathogenesis of vitiligo. On page 6-7 of the article, second paragraph, lines 128-143.

Comments 2: Lines 40-42; 61-65: references are needed

Response 2: Thank you for your careful review, and we have added the relevant reference in lines 40-42; 61-65 of the manuscript as you suggested. In the revised manuscript, page 5, first paragraph, lines 91.

Comments 3: Line 99: ARE regions are located in the promoters of genes coding for antioxidant enzymes

Response 3: Thank you for your careful review, we have revised the corresponding descriptions in the manuscript in accordance with your comments. In the revised manuscript, page 8, second paragraph, line 178.

Comments 4: Figure 1C: Images are too small

Response 4: Thank you for your suggestion, we have modified the size and resolution of the corresponding images according to your suggestion.

Comments 5: Figure 2A: are needed

Response 5: Thank you for pointing out our error, we have added scale bars to the corresponding images as per your comments.

Comments 6: Figure 3F and 4A: images are too small to appreciate the staining localization

Response 6: Thank you for your suggestion, we have added enlarged partial images and added scale bars to the corresponding images as per your comments.

Comments 7: Why authors did not use normal human melanocyte? 

Response 7: The main reasons for using B16F10 cells include the following reasons:

1. Ease of culture and manipulation: B16F10 cells are a type of mouse melanoma cells that are characterised by their fast growth rate, ease of passaging and relatively low requirements for culture conditions, which makes them the cell model of choice for screening for whitening or tanning-promoting active ingredients.

2. Large-scale or high-throughput screening: Due to their easy culture characteristics, B16F10 cells are widely used for large-scale or high-throughput screening of whitening or tanning active ingredients, in order to rapidly assess the whitening or tanning effects of different ingredients.

3. In vitro model: The B16F10 cell model is an in vitro model that can simulate the melanogenesis process in human skin to assess the effect of whitening or tanning ingredients on melanogenesis, which is very useful for studying the efficacy of whitening or tanning products.

4. Cost-effectiveness: Compared to using human skin samples or 3D skin models, experiments using B16F10 cells are less costly and large numbers of cells can be obtained in a short period of time, which is suitable for conducting a large number of experimental studies.

5. Wide research base: B16F10 cells have been widely used in whitening or tanning-related research, and researchers have a deeper understanding of their biological properties and response mechanisms, which helps to better understand and analyse the results of experiments.

In summary, B16F10 cells are widely used in whitening-related research due to their advantages of easy cultivation, cost-effectiveness, and broad research base.

Reference[1-6]

1.     Sim MO, Ham JR, Lee MK. Young leaves of reed (Phragmites communis) suppress melanogenesis and oxidative stress in B16F10 melanoma cells. Biomed Pharmacother. 2017 Sep;93:165-71.

2.     An X, Lv J, Wang F. Pterostilbene inhibits melanogenesis, melanocyte dendricity and melanosome transport through cAMP/PKA/CREB pathway. Eur J Pharmacol. 2022 Oct 15;932:175231.

3.     Uto T, Tung NH, Ohta T, Shoyama Y. (+)-Magnolin Enhances Melanogenesis in Melanoma Cells and Three-Dimensional Human Skin Equivalent; Involvement of PKA and p38 MAPK Signaling Pathways. Planta Med. 2022 Oct;88(13):1199-208.

4.     Lv J, Fu Y, Gao R, Li J, Kang M, Song G, et al. Diazepam enhances melanogenesis, melanocyte dendricity and melanosome transport via the PBR/cAMP/PKA pathway. Int J Biochem Cell Biol. 2019 Nov;116:105620.

5.     Jeong HS, Gu GE, Jo AR, Bang JS, Yun HY, Baek KJ, et al. Baicalin-induced Akt activation decreases melanogenesis through downregulation of microphthalmia-associated transcription factor and tyrosinase. Eur J Pharmacol. 2015 Aug 15;761:19-27.

6.     Zheng Y, Lee EH, Lee SY, Lee Y, Shin KO, Park K, et al. Morus alba L. root decreases melanin synthesis via sphingosine-1-phosphate signaling in B16F10 cells. J Ethnopharmacol. 2023 Jan 30;301:115848.

Comments 8: 4.1. Materials: I suggest to move the antibodies used in a dedicate table 

Response 8: Thanks to your suggestion, the antibodies used in the 4.1 material are written with the corresponding article numbers as well as the dilution ratios.

Comments 9: Authors must report the product code of all kits used

Response 9: Thank you very much for your careful attention. We have written the CAT. Number on the back of the corresponding kits.

Comments 10: Abbreviations must be written in full length when mentioned for the first time

Response 10: Thank you very much for your suggestion. The full names of the corresponding abbreviations are already listed in the ‘Abbreviations’ section of the manuscript, but probably because the file generated after submission does not show the contents of the abbreviations, we have now included the abbreviations as follows.

Abbreviations

Akt, protein kinase (PKB); BNIP3L/Nix, Bcl-2/adenovirus E1B 19 kDa protein-interacting protein 3-like; Cdc42, Cell division cycle 42; DCFH-DA, 2,7-Dichlorod -hydrofluorescein diacetate; DRP1, dynamin-related protein 1; FBS, Fetal bovine serum; GAPDH, Glyceraldehy phosphate dehydrogenase; HO-1, Heme Oxygenase-1; Keap1, Recombinant Kelch Like ECH Associated Protein 1; MFF, Mitochondrial Fission Factor; MDA, Malondialdehyde; MTT, Methylthiazolyl tertrazolium bormide; MMP, Mitochondrial membrane potential; mTOR, mammalian target of Rapamycin; NQO1, NAD(P)H: quinone oxidoreductase 1; Nrf2, Nuclear factor-E2-related factor 2; PAGE, Polyacrylamide gel electrophoresis; PVDF, Polyvinylidene Fluoride; PBS, Phosphate buffered saline; PMSF, Phenylmethl sulfonylfluoride; PI3K, phosphatidylinositol 3-kinase; Rac-1, Ras-related C3 botulinum toxin substrate 1; ROS, Reactive oxygen species; SDS, Sodium didecylsulfate; SOD, Superoxide dismutase; TRP1, tyrosinase-related protein 1; TRP2, dopachrome tautomerase; TYR, Tyrosinase.

Comments 11: Comments on the Quality of English Language

An accurate revision of syntax is recommended

Response 11:

Thank you for your seriousness and responsibility. We have carefully checked the grammar in the manuscript according to your suggestions, and if there are still any errors, we hope you can tell us in detail what exactly needs to be changed!

Once again, we sincerely thank you for your suggestions on this manuscript, and if there are still deficiencies after the revision, we hope you can give us your valuable suggestions, we will be grateful!

Round 2

Reviewer 1 Report

Comments and Suggestions for Authors

Authors have respond to majority of comments, however still some few corrections are required before publication:

1.     Title is concise it should be more representable of the study work taking in consideration both in vivo and in vitro studies

2.     Please revise numbering values in the Results section lines 153, 181, 205, 213, 252, 265, 288, 304

3.     Lines 148, 159, 184, 186, 224, 241, 244,  254, 269, 293 297, 307,  : citing references not in the results section, please move to the section of discussion

4.     All sub-headings in the result section should be modified as the following. for example in the sub-heading (line 252) it could be modified from: Apigenin promoted mitochondrial fusion while suppressed mitochondrial fission to: the effect of Apigenin on mitochondrial fusion and mitochondrial fission. Similarly lines 265, 288, 304

5.     For each assay used what is the reference of such method please cite reference for each method used in the experimental measurements

Figure 1 is overcrowded that make it difficult to be clarified please modify as figure contains several panels A,B, C, D, E, F, G, H, I, J , and K thus it is overcrowded as many panels are not clear. Please divide into 2 figures for better representation of your results

Comments on the Quality of English Language

the grammatical and language check must be done in a certified manner as no corrections were done by authors and differs from the first version of the manuscript.

Author Response

For research article

Response to Reviewer 1 Comments

1. Summary

2. Questions for General Evaluation

Reviewer’s Evaluation

Response and Revisions

Does the introduction provide sufficient background and include all relevant references?

Yes

Thank you for your careful review, and we have revised the manuscript accordingly, following your suggestions.

Are all the cited references relevant to the research?

Yes

Thank you for your careful review, and we have revised the manuscript accordingly, following your suggestions.

Is the research design appropriate?

Yes

Thank you for your careful review, and we have revised the manuscript accordingly, following your suggestions.

Are the methods adequately described?

Can be improved

Thank you for your careful review, and we have revised the manuscript accordingly, following your suggestions.

Are the results clearly presented?

Can be improved

Thank you for your careful review, and we have revised the manuscript accordingly, following your suggestions.

Are the conclusions supported by the results?

Yes

Thank you for your careful review, and we have revised the manuscript accordingly, following your suggestions.

3. Point-by-point response to Comments and Suggestions for Authors

Comments 1: Title is concise it should be more representable of the study work taking in consideration both in vivo and in vitro studies

Response 1: Thank you for pointing this out. We agree with this observation. Therefore, we have revised the article title to Apigenin ameliorates H2O2-induced oxidative damage in melanocytes through Nrf2 and PI3K/Akt/mTOR pathways and reducing the generation of ROS in zebrafish. Thank you again for your suggestion, and we hope that you will read the revised article title, and if it is not appropriate, we hope that you will make your suggestion and we will revise it.

Comments 2: Please revise numbering values in the Results section lines 153, 181, 205, 213, 252, 265, 288, 304

Response 2: Thank you for your careful review, numbering values in the results section lines  153, 181, 205, 213, 252, 265, 288, 304 in the manuscript is correct, the PDF file generated by the submission system are 2.1, we do not know how to correct, we are very sorry for the inconvenience caused to you.

Comments 3: Lines 148, 159, 184, 186, 224, 241, 244,  254, 269, 293 297, 307,  : citing references not in the results section, please move to the section of discussion

Response 3: Thank you for your careful review, lines 148, 159, 184, 186, 224, 241, 244,  254, 269, 293 297, 307,  : citing references not in the results section, we have moved to the discussion section.

Comments 4: All sub-headings in the result section should be modified as the following. for example in the sub-heading (line 252) it could be modified from: Apigenin promoted mitochondrial fusion while suppressed mitochondrial fission to: the effect of Apigenin on mitochondrial fusion and mitochondrial fission. Similarly lines 265, 288, 304

Response 4: Thank you very much for your suggestion, We have modified the titles of the corresponding sections as you suggested, highlighted in red in the manuscript. If there is still a need for revision, please let us know and we will improve it according to your suggestions.

Comments 5: For each assay used what is the reference of such method please cite reference for each method used in the experimental measurements

Response 5: Thank you for your suggestion, we have cited reference for each method used in the experimental measurements. On page 24 of the manuscript, third paragraph, line 514, page 25 of the manuscript, first and second paragraphs, lines 523 and 532, page 26 of the manuscript, first, second and third paragraphs, lines 545, 553 and 565.

Comments 6: Figure 1 is overcrowded that make it difficult to be clarified please modify as figure contains several panels A,B, C, D, E, F, G, H, I, J , and K thus it is overcrowded as many panels are not clear. Please divide into 2 figures for better representation of your results

Response 6: Thank you very much for your suggestion, and based on your suggestions, we have divided Figure 1 into 2 figures. Thanks again for your careful review.

Comments 7: The grammatical and language check must be done in a certified manner as no corrections were done by authors and differs from the first version of the manuscript.

Response 7: Thank you for your seriousness and responsibility. We have carefully checked the grammar in the manuscript according to your suggestions, and if there are still any errors, we hope you can tell us in detail what exactly needs to be changed!

Once again, we sincerely thank you for your suggestions on this manuscript, and if there are still deficiencies after the revision, we hope you can give us your valuable suggestions, we will be grateful!

Reviewer 2 Report

Comments and Suggestions for Authors

The authors have made the requested changes precisely and appropriately to thus improve the work. The manuscript can be accepted for publication in the present form. 

Author Response

For research article

Response to Reviewer 2 Comments

1. Summary

Thank you very much for taking the time to review this manuscript.

2. Questions for General Evaluation

Reviewer’s Evaluation

Response and Revisions

Does the introduction provide sufficient background and include all relevant references?

Yes

Are all the cited references relevant to the research?

Yes

Is the research design appropriate?

Yes

Are the methods adequately described?

Yes

Are the results clearly presented?

Yes

Are the conclusions supported by the results?

Yes

Once again, we sincerely thank you for your valuable comments on this manuscript and for your careful review! We wish you a happy life and good luck in your work!

Reviewer 3 Report

Comments and Suggestions for Authors

the manuscript has been significantly improved and can be accepted in the present form 

Author Response

(The authors gave the same response as above.)
